# Blood Speckle Imaging: An Emerging Method for Perioperative Evaluation of Subaortic and Aortic Valvar Repair

**DOI:** 10.3390/bioengineering10101183

**Published:** 2023-10-12

**Authors:** Elias Sundström, Michael Jiang, Hani K. Najm, Justin T. Tretter

**Affiliations:** 1Department of Engineering Mechanics, FLOW Research Center, KTH Royal Institute of Technology, Teknikringen 8, 100 44 Stockholm, Sweden; 2Department of Pediatric Cardiology, Cleveland Clinic, Cleveland, OH 44195, USA; 3Congenital Valve Procedural Planning Center, Department of Pediatric Cardiology, Cleveland, OH 44195, USA; 4Division of Pediatric Cardiac Surgery, and the Heart, Vascular, and Thoracic Institute, Cleveland Clinic, Cleveland, OH 44195, USA

**Keywords:** blood speckle imaging, aortic valve repair, aortic stenosis

## Abstract

Background: This article presents the use of blood speckle Imaging (BSI) as an echocardiographic approach for the pre- and post-operative evaluation of subaortic membrane resection and aortic valve repair. Method: BSI, employing block-matching algorithms, provided detailed visualization of flow patterns and quantification of parameters from ultrasound data. The 9-year-old patient underwent subaortic membrane resection and peeling extensions of the membrane from under the ventricular-facing surface of all three aortic valve leaflets. Result: Post-operatively, BSI demonstrated improvements in hemodynamic patterns, where quantified changes in flow velocities showed no signs of stenosis and trivial regurgitation. The asymmetric jet with a shear layer and flow reversal on the posterior aspect of the aorta was corrected resulting in reduced wall shear stress on the anterior aspect and reduced oscillatory shear index, which is considered a contributing element in cellular alterations in the structure of the aortic wall. Conclusion: This proof-of-concept study demonstrates the potential of BSI as an emerging echocardiographic approach for evaluating subaortic and aortic valvar repair. BSI enhances the quantitative evaluation of the left ventricular outflow tract of immediate surgical outcomes beyond traditional echocardiographic parameters and aids in post-operative decision-making. However, larger studies are needed to validate these findings and establish standardized protocols for clinical implementation.

## 1. Introduction

Aortic valvar repair is increasingly pursued for treating aortic valvar pathologies when feasible, aiming to restore valvar function and avoid the need for valvar replacement. This, however, requires a thorough understanding of the complex anatomy of the aortic root and its valve, and technical proficiency which is only obtained with experience and time spent on this steep learning curve [1]. Accurate evaluation of the repaired valve is crucial for assessing immediate surgical outcomes and optimizing post-operative management. While traditional echocardiographic techniques offer valuable information, they may have limitations in capturing detailed hemodynamic changes associated with aortic valvar repair [2]. On many occasions, we lack understanding of the recurrence of the subaortic obstruction, despite a good initial gradient drop. This lack of proper assessment of flow dynamics may explain the reason for the recurrence of the subaortic membrane and moderate progressive subaortic stenosis.

Continuous-wave Doppler echocardiography is a standard clinical procedure for evaluating the extent of aortic stenosis. It involves measuring the peak velocity of blood flow as it passes through the aortic valve during systole. By applying the simplified Bernoulli equation, clinicians can estimate the transvalvar pressure gradient [3]. This non-invasive technique is preferred over cardiac catheterization due to its accessibility, affordability, and minimal invasiveness [4].

The application of continuous-wave Doppler echocardiography has been found to be limited to the estimation of peak velocity and trasvalvar pressure gradient when contrasted with the equation that factors in the complete hemodynamic profile at the point of greatest constriction [5,6]. Relying solely on peak velocity measurements neglects the momentum of blood flow spanning the entire vascular cross-section, a crucial element in accurately estimating the velocity profile and Wall Shear Stress (WSS). Moreover, the estimation of peak velocity using Doppler echocardiography heavily relies on the skills of the operator. Any misalignment between the angle of insonation and the direction of blood flow can result in inaccuracies of the maximum velocity [7]. Numerous non-invasive alternatives have been investigated, although they have not yet been incorporated into clinical practice [6,8].

Blood speckle imaging (BSI) has emerged as a recent alternative approach for evaluating the severity of aortic stenosis [9,10]. This technique involves the direct measurement and visualization of blood vector velocity patterns, captured at ultra-high frame rates in the kilohertz range [9,11,12]. BSI holds promise in mitigating the limitations of conventional Doppler echocardiography, such as angle dependence and the reliance on acquiring single peak velocities [5].

BSI relies on existing technology for tissue speckle tracking, applied to assess myocardial deformation [13]. The methodology entails defining a small image kernel within the initial vessel image, then tracking the same speckle pattern in subsequent frames using a “best match” search algorithm. This process is iterated across a grid of measurements to quantify both the velocity and direction of blood flow [9,14]. This approach to obtaining blood flow velocity data offers an advantage, potentially enabling the calculation of shear stress from velocity information across a cross-sectional profile, as opposed to relying on a single streamline as seen in traditional Doppler echocardiography.

This proof-of-concept study explores the retrospective application of BSI as a novel echocardiographic approach for pre- and post-operative evaluation of aortic valve repair in one 9-year-old patient with recurrent subaortic membrane. Tissue and hemodynamics data were acquired with the 9VT-D transesophageal probe by GE HealthCare (GE Vingmed Ultrasound AS, Horten, Norway), which is the first 4D transesohpageal echocardiography transducer small enough for use in young children. To the authors’ knowledge, this is the first paper describing its use with BSI. This echocardiographic imaging mode acquires 2D images at an exceptionally fast temporal frequency and employs a block-matching algorithm [15,16,17,18,19,20] to track the speckle movement across the images to produce 2D vector fields. The flow field visualization allows for further characterization of the flow dynamics including turbulence and shear force calculation [21,22,23,24]. We sought to utilize BSI data in this clinical case to retrospectively analyze the flow patterns and derived properties to visually assess and quantify the hemodynamic changes from surgical subaortic membrane resection and aortic valvar repair.

## 2. Methods

### 2.1. Surgical Repair

The 9-year-old patient with recurrent subaortic membranes underwent surgical resection in accordance with standard clinical practice. Deliberate interrogation of the leaflet revealed the non-obvious extension of the subaortic membrane onto the leaflets. This membranous extension was delicately peeled off, uncovering a thin pliable leaflet under the inferior (ventricle-facing) surface. Standard intra-operative transesophageal imaging demonstrated satisfactory resection of subaortic membrane without significant aortic valvar regurgitation or stenosis.

### 2.2. Quantification of the Velocity Field from BSI

The Vivid E95 ultrasound system and 9VT-D transesophageal probe by GE HealthCare (GE Vingmed Ultrasound AS, Horten, Norway) were used for the acquisition of the BSI images. The BSI images consisted of a cine of standard 2D sonographic images, blood velocity data, confidence level, and metadata. The tissue data consisted of single channel data, where the pixel value represents the image brightness, see Figure 1a. The velocity data were multichannel and were combined into a velocity field. A second single channel was used for quantification of the confidence level in the blood velocity, with a range of 0 as low and 1 as high confidence [25]. A confidence level of 1 corresponded to a high fidelity of the block-matching protocol that tracks the displacement of blood speckles. Low confidence levels indicate less reliable velocity estimates and could be caused by low image quality or blood speckles moving out of plane [22,23,24].

The physical distance between pixels in the BSI images defines the spatial extent of the image and allows determining the size of each pixel, which was 0.0825 mm/pixel. The number of pixels in the x- and y-directions were obtained from the tissue data, which were subsequently used for creating a 2D grid spanning the region of interest. The velocity field was then mapped on this 2D grid to facilitate visualization and flow quantification; see Figure 1b,c. The BSI images were acquired with a pulse repetition frequency (PRF) of 6 kHz and with a maximum velocity of 2 m/s of the tracking in both pre- and post-operative cases.

### 2.3. Quantification of Wall Shear Stress Indicators

Wall shear stress (WSS) is an important parameter to characterize the force per unit area that the blood flow exerts on the endothelial wall in the aorta. WSS varies in space and time depending on the blood flow conditions and geometry of the thoracic aorta and the aortic valve. To characterize WSS in the preoperative and postoperative cases, the study used two common parameters. TAWSS, or time-averaged wall shear stress, represents the localized, time-averaged value of wall shear stress (WSS) [26,27].
(1)$$TAWSS=\frac{1}{T}\int_{0}^{T}|WSS_{i}|dt.$$

Spatial variations in WSS provide insights into the magnitude and the non-uniformity of WSS, but they do not convey information about the temporal variations in WSS. Therefore, the study also uses the oscillatory shear index (OSI) for quantification of frequency fluctuations in the wall shear stress (WSS) [27], which is defined as
(2)$$OSI=\frac{1}{2}\left(1-\frac{|\int_{0}^{T}WSS_{i}dt|}{\int_{0}^{T}|WSS_{i}|dt}\right).$$

OSI is a parameter that ranges from 0 to 0.5. When WSS remains consistently positive (whether oscillatory or non-oscillatory), OSI equals 0. When the WSS changes sign to the extent that the integral of the positive and negative sequences becomes equal, OSI reaches the value of 0.5.

## 3. Results

### 3.1. Echocardiographic Improvement after Subaortic Membrane Resection and Aortic Valvar Repair

During the pre-operative transesophageal echocardiogram (Figure 2), BSI depicted abnormal flow patterns and high peak velocity associated with the left ventricular outflow obstruction and restricted valvar leaflets. Figure 3 demonstrates significant improvements in flow patterns, with quantified changes in flow velocities indicating the absence of stenosis and minimal regurgitant volumes.

### 3.2. BSI Confidence Level

Figure 4 shows the confidence level for the pre- and post-operative cases during the cardiac cycle. The post-operative case shows high confidence levels in the region around the aortic root and proximally of the sinutubular junction between peak systole t/T=0.2 and towards the end of systole t/T=0.4, where *t* is time and *T* is the period of the cardiac cycle. The pre-operative case shows less uniformity with local spots of low confidence in the aortic root, especially around the stenotic left ventricular outflow tract and the aortic valve. Both cases indicate lower confidence in the left atrium (LA), which is at a depth further from the TEE transducer.

### 3.3. BSI Flow Field

Figure 5 shows the blood flow through a parasternal long-axis view of the left ventricular outflow tract and aortic valve during different stages of the cardiac cycle for both the pre- and post-operative cases. In the early systole or isovolumetric contraction t/T=0, the aortic valve is closed with minimal flow velocity. At early and late systole t/T=0.2 and 0.4, respectively, the valve is open, and the velocity vector field shows flow acceleration across the aortic valve. This flow acceleration in the pre-operative case appears more prominent and is directed slightly anteriorly (upwards image). In late systole, a large recirculation zone forms in the posterior region of the aortic root (downstream from the stenotic aortic valve). In contrast, the post-operative vector field demonstrates a smoother flow profile throughout the systole.

At the end of the systolic phase, the valve closes, and the velocity magnitude of the blood flow reduces, indicating a decrease in aortic pressure during diastole. At the time of mid-diastole t/T=0.8, following isovolumetric relaxation, the mitral valve opens, discharging blood flow from the left atrium to the left ventricle, i.e., filling of the left ventricle to prepare for a new systole.

### 3.4. BSI Peak Velocity Compared with CW Doppler Measurements

Figure 6 shows the continuous-wave (CW) Doppler assessment for the pre-operative and the post-operative transesophageal echocardiographic studies surrounding the surgical procedure. There is a peak velocity of approximately 3.5 m/s at peak systole for the pre-operative case, indicating moderate left ventricular outflow tract and aortic valvar stenosis. In the post-operative case, the maximum velocity at peak systole is around 1.4 m/s, indicating resolution of the left ventricular outflow tract and aortic valve obstruction.

The maximum velocity at the narrow section of the valve as obtained with BSI and CW Doppler for both pre-operative and post-operative cases are compared in Figure 7a. In the pre-operative case, there is a larger difference in maximum velocity at peak systole, where BSI indicates lower levels. This coincides with the lower BSI confidence level in the pre-operative case; see Figure 7b. In the post-operative case, there is a good agreement, both in terms of peak velocity and its variation during the cardiac cycle.

### 3.5. BSI Velocity and Shear Stress Quantification

The blood flow presented in Figure 5 is further quantified along the vertical profile (white dashed line in Figure 5). In the pre-operative case, the long-axis velocity component indicates low magnitudes at early systole t/T=0, which progress towards peak systole t/T=0.2 and forming an anteriorly directed jet with flow reversal on the posterior aspect of the aortic root. There is a larger velocity gradient at the shear layer of the jet, which results in increased shear stress (bottom left panel). Towards post-systole t/T=0.4, the velocity magnitude decreases and the center of the jet is displaced approximately 5 mm in the short axis direction. In the post-operative case (right top panel in Figure 8), a relatively symmetric top-hat velocity profile develops at peak systole, with only minor recirculation towards the endothelial walls. In this case, there is a steeper gradient in the jet’s shear layer on the side posteriorly of the probe, resulting in increased shear stress level (bottom right panel in Figure 8). There is also less displacement along the short axis of the jet’s center in the post-operative case compared with the pre-operative case.

In both pre-operative and post-operative cases, the axial velocity undergoes a change in sign, transitioning from positive to negative flow, which coincides with the location of the shear layer. In the preoperative case, the retrograde flow towards the posterior wall is stronger than in the postoperative case. Nevertheless, integrating over the cardiac cycle shows that the peaks of the time average wall shear stress (TAWSS) are similar, i.e., between 1.5 and 1.6 for both cases; see Table 1. However, a retrograde flow with a change in flow direction results in a non-zero oscillatory shear index in the axial flow direction [26,27]. This is indicated in Table 1 where the OSI is significantly larger in the pre-operative case. Time durations with significant wall shear stress in combination with non-zero OSI have been suggested as a risk factor for cell-driven changes in the aortic wall structure [28], including development and progression of atherosclerosis [29,30].

## 4. Discussion

In this study, we used blood speckle Imaging for pre- and post-operative evaluation of subaortic membrane resection and aortic valvar repair in a young pediatric patient utilizing the newly available mini-3D transesophageal probe. The post-operative case showed high BSI confidence levels in the region of the aortic root, whereas the pre-operative case showed relatively lower and less uniform BSI confidence levels in this region. The suboptimal BSI confidence level obtained pre-operatively resulted in an underprediction of peak velocities compared to the CW Doppler measurement. In cases with suboptimal BSI confidence levels, it would be necessary to resort to CW Doppler assessment of peak velocity.

Despite the limitation with BSI and its measurement of peak velocity in the pre-operative case, it is possible to quantify its velocity profile and compare it with the post-operative case, which is not possible with CW Doppler. This comparison revealed that the pre-operative case exhibits a more anteriorly directed jet with a large recirculation zone on the posterior side of the aortic root. In the post-operative case at peak systole, the axial velocity data in the ascending aorta demonstrated a more symmetric top-hat profile, which is in good agreement with previous observations [31,32,33,34,35]. The steeper velocity gradient led to a higher axial shear stress in the shear layer in the post-operative case. However, this is most likely due to the underestimation of the peak velocities with BSI in the stenotic pre-operative case.

Both pre-operative and post-operative cases indicate a non-zero oscillator shear index (OSI), since the axial velocity profile undergoes a change in sign, i.e., alternating from positive to negative flow during the cardiac cycle [26,27]. This is more significant in the pre-operative case due to its stronger recirculation zone that is located towards the posterior side of the aortic valve. In the study by Kiema et al. (2022) [28], it was observed that in cases of a situation of non-zero OSI together with significant WSS, there is elevated risk for the incidence of aortic wall damage, including progression of atherosclerosis [29,30].

The quantitative assessment of axial velocity and shear stress contributes to a deeper understanding of the relationship between hemodynamics in the proximal thoracic aorta and differences in aortic valvar morphology. The Computational Fluid Dynamics model in patient-specific geometries [36,37,38] will be performed to compare with the observed behavior. Despite BSI’s current capabilities constraining velocity magnitude to 2 m/s and within 2D imaging planes, it may serve as a powerful tool to validate any simulation results. In areas of laminar blood flow without extreme velocities, such as the primarily regurgitant aortic valve, the 3D vector field generated from BSI may serve as an inlet boundary condition for computational models. Furthermore, even in the aortic valve with significant stenosis or subvalvar obstruction, post-operative assessment following repair may help to assess immediate hemodynamic results and predict long-term valvar repair durability. While the recurrence of a subaortic membrane may be inevitable in certain cases, this technique may provide a better understanding of the completeness of the surgical intervention when compared to simple Doppler and color Doppler assessment of the residual gradient or leak. In addition, this may add some early indications for the substrate for possible recurrence in the future. Further investigations involving larger cohort sizes will be conducted to determine the potential clinical utility of these findings, particularly in relation to assessing for favorable hemodynamics following left ventricular outflow tract and aortic valvar repair.

## 5. Limitations

In the pre-operative case, for a considerable duration of systole, the flow acceleration through the stenotic subaortic membrane and the valve exceeds the upper limit of BSI of 2 m/s. The BSI block-matching algorithm is unable to track the speckle pattern beyond the proportional maximum search distance, resulting in unpredictable underestimation of the flow velocity. In the post-operative case, with velocities never exceeding the 2 m/s limit, the agreement as compared to CW Doppler is good, both in terms of magnitude and its variation during the cardiac cycle. Thus, BSI demonstrates robustness when applied to the aortic flow with moderate flow velocities.

Another limitation of the BSI employed in this study is that it is limited to 2D planar assessment. The velocity component orthogonal to the imaging plane is not available. Capturing the maximum velocity can be optimized by aligning it with the long axis of the LVOT and aorta. However, we lose the vorticity around this long-axis, as well as any jet or turbulent flow which may deviate outside the 2D imaging plane. The 3D physiological displacement of the heart throughout the cardiac cycle, which evolves both spatially and temporally, also is not fully captured. This influences the flow quantification of relative velocities to the walls in the left ventricle, as it experiences significant volume changes during systolic contraction. However, for vessels that undergo moderate displacement and area changes, this factor is less significant. Nonetheless, 4D BSI would help overcome this limitation.

There is a need to validate both BSI and CW Doppler techniques in the setting of multilevel obstruction. Previous studies have measured the pressure drop in silicon phantom models of the aortic valve using PendoTech pressure sensors to assess the accuracy of CW Doppler and BSI [6]. However, so far, it has been challenging to perform these pressure measurements in vivo.

## 6. Conclusions

For the postoperative non-stenotic condition, blood speckle imaging (BSI) demonstrated good agreement with the continuous wave Doppler echocardiography estimation of peak velocity. This demonstrates the potential for assessing shear stresses within the aortic valve region. However, BSI shows a tendency to underestimate peak velocities for the preoperative condition, with high stenotic burden, likely due to limitations in tracking higher flow velocities and managing speckle decorrelation. Despite similar TAWSS in both cases, it was observed that the OSI level reduced significantly postoperatively, indicating a lower risk factor for the ongoing development and progression of adjacent aortic wall and leaflet damage, including atherosclerosis.

While BSI presents several advantages over traditional Doppler methods, its use for shear stress assessments in aortic stenosis needs further capability. Extended clinical investigations are imperative before BSI can be considered a viable means to enhance the accuracy of wall shear stress estimations in the clinical assessment of aortic stenosis.

## Figures and Tables

**Figure 1 bioengineering-10-01183-f001:**
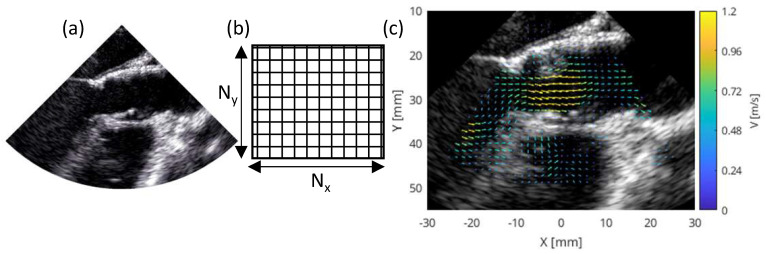
Blood speckle imaging (BSI) data (**a**) consisting of tissue (single channel) and velocity (multi-channel). The multichannel data were combined into a velocity field and then mapped onto the 2D grid (**b**) for visualization and quantification of the flow (**c**).

**Figure 2 bioengineering-10-01183-f002:**
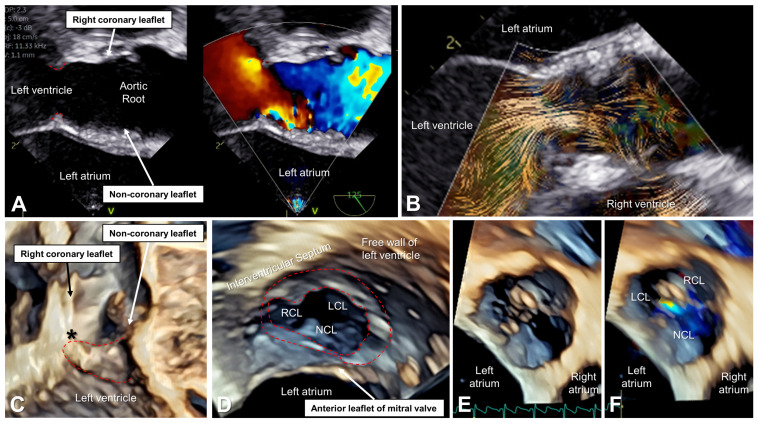
Pre-operative transesophageal echocardiographic evaluation. (**A**) Color Doppler comparison in the systole demonstrates flow acceleration starting at the level of the subaortic membrane (the membrane is outlined with red hashed lines in this and the other panels). (**B**) BSI in the systole demonstrates narrowing of the outflow jet at the level of the subaortic membrane with subsequent turbulence. (**C**) 3D long-axis imaging in the systole shows the prominent subaortic membrane which has an attachment to the nadir of the right coronary leaflet (black asterisk) with mild thickening of the leaflet extending from the leaflet hinge to the tip. (**D**) 3D imaging in the systole viewing in the short axis from the apex of the ventricle looking up towards the left ventricular outflow tract demonstrates the circumferential membrane which is most prominent under the coronary leaflets. (**E**) 3D short axis of the aortic valve in the diastole (**F**) with a 3D color Doppler demonstrates the trileaflet aortic valve with notable thickening of the right coronary leaflet (RCL) with mild to moderate central regurgitation. LCL, left coronary leaflet; NCL, non-coronary leaflet.

**Figure 3 bioengineering-10-01183-f003:**
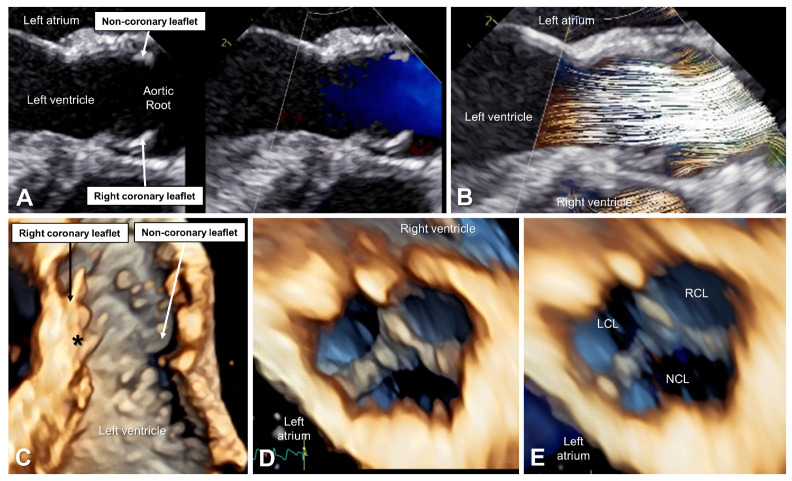
Post-operative transesophageal echocardiographic evaluation. (**A**) Color Doppler comparison in the systole demonstrates no significant residual subaortic membrane with laminar flow across the left ventricular outflow tract and the aortic valve. (**B**) BSI in the systole confirms laminar flow. (**C**) 3D long-axis imaging in the systole shows a very trivial residual subaortic membrane near the nadir of the right coronary leaflet (black asterisk) with a noticeably thinner right coronary leaflet. (**D**) 3D short axis of the aortic valve in the diastole (**E**) with a 3D color Doppler demonstrates notable thinning of the right coronary leaflet (RCL) tip with improve coaptation and no significant residual regurgitation. LCL, left coronary leaflet; NCL, non-coronary leaflet.

**Figure 4 bioengineering-10-01183-f004:**
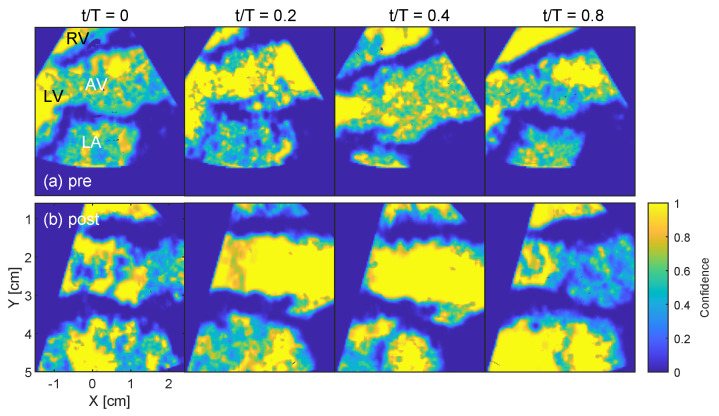
Parasternal long-axis view of the left ventricular (LV) outflow tract and the aortic valve (AV) showing the BSI confidence level, with range 0 as low and 1 as high confidence. The TEE probe has the same orientation as in Figure 1 and the confidence level is shown at the beginning of systole t/T=0, peak systole t/T=0.2, towards the end of systole t/T=0.4, and during diastole t/T=0.8. Pre-operative images are displayed across the top panels and post-operative images across the bottom panels.

**Figure 5 bioengineering-10-01183-f005:**
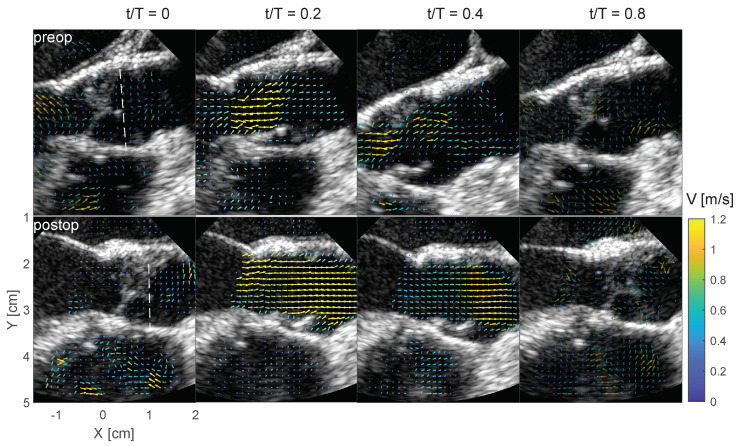
Parasternal long-axis view of the left ventricular (LV) outflow tract and the aortic valve (AV) showing the velocity field at different time instants during the cardiac cycle, where the vector color indicates the velocity magnitude. The TEE probe has the same orientation as in Figure 1. Pre-operative images are displayed across the top panels and post-operative images are displayed across the bottom panels.

**Figure 6 bioengineering-10-01183-f006:**
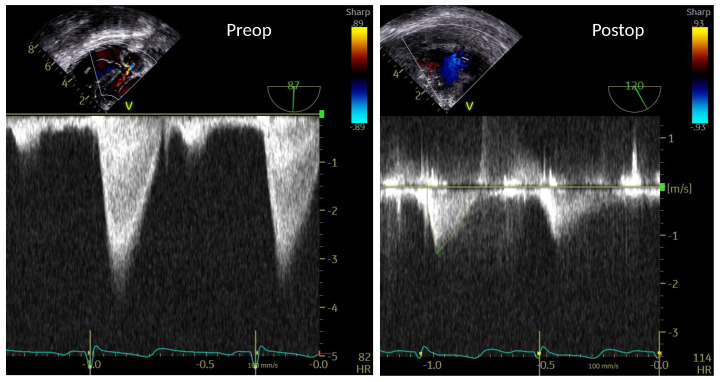
Continuous-wave Doppler assessment from the pre-operative (**left panel**) and post-operative (**right panel**) transesophageal echocardiographic studies.

**Figure 7 bioengineering-10-01183-f007:**
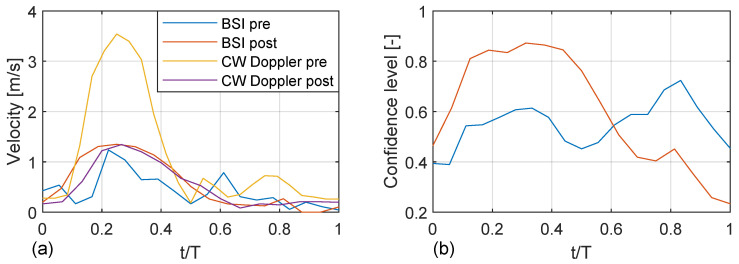
(**a**) Maximum velocity at peak systole as a function of the cardiac cycle for pre- and post-operative cases, for BSI and continuous-wave (CW) Doppler data, respectively. The maximum velocity is evaluated as the maximum valve on the white dashed lines in Figure 5. (**b**) Confidence level as a function of the cardiac cycle for the pre- and post-operative cases (evaluated as the mean value on the white dashed lines in Figure 5).

**Figure 8 bioengineering-10-01183-f008:**
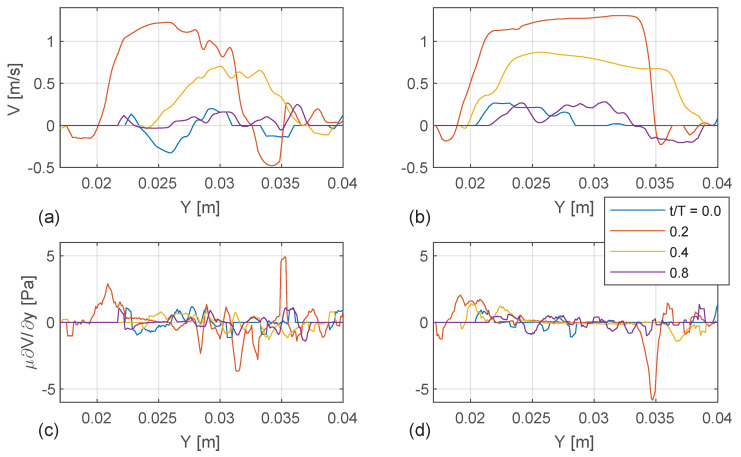
The long-axis velocity component (V) is given as a function of the short-axis (Y) for (**a**) pre-operative and (**b**) post-operative cases. The long-axis shear stress (μ∂V/∂y) component is given on the bottom row for (**c**) pre-operative and (**d**) post-operative cases. Both the velocity and the shear stress are given at different instances of the cardiac cycle (t/T=0,0.2,0.4,0.8). The location of the profiles is along the white dashed line, annotated in Figure 5. The y-direction corresponds with the short axis, where a lower y is towards the anterior wall (higher up in the screen and closer to the probe, c.f. Figure 2 and Figure 3) and a higher y is towards the posterior wall (more distal to the probe).

**Table 1 bioengineering-10-01183-t001:** Peak values of WSS indicators between pre-operative and post-operative cases.

Case	Pre-Operative	Post-Operative
TAWSS	1.52	1.58
OSI	0.24	0.01

## Data Availability

Data available on request due to restrictions in repository access.

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
