# Peer review of "Blood Speckle Imaging: An Emerging Method for Perioperative Evaluation of Subaortic and Aortic Valvar Repair"

_bioengineering, 2023, doi:10.3390/bioengineering10101183_

Round 1

Reviewer 1 Report

This article presents the use of Blood Speckle Imaging (BSI) as a novel echocardiographic approach for the pre- and post-operative evaluation of subaortic membrane resection and aortic valve repair. The authors believe that this is the first paper describing the use of tissue and hemo-dynamics data with BSI. Sundström and co-authors sought to utilize BSI data in their clinical case to retrospectively analyze the flow patterns and derived properties to visually assess and quantify the hemodynamic changes from surgical subaortic membrane resection and aortic valvar repair.

Major:

1.   The authors propose a new application scenario, but do not propose an innovation in principle. In particular, there is no specific principle, if necessary, please list specific formulas and specific detection mechanisms.

2.   Similar to laser Doppler and diffused light correlation spectroscopy measurement techniques, laser speckle contrast imaging methods are based on the basic principle of dynamic light scattering, and various approximate models are used to approximate the autocorrelation function of speckle to obtain the rate of moving scattering particles. Doppler technology is listed in the manuscript, but there is nonadequately comparison, what are the advantages and disadvantages of BSI technology?

3.  The legends in Figures 7 and 8 are vague, and it is recommended to add a separate legend to each image.

4.  The structure of the manuscript is a bit strange, section 4 has only Discussion sections, and the Discussion section is too long. It should be divided into two sections, and a Conclusion section should be added.

Minor:

1.   Section 2.1 is too short and should be merged into other chapters.

2.  The speckle lining result correlates strongly with the flow rate of speckle particles per unit time, that is, the average velocity. In line 109, the peak speed is shown, if expressed in terms of average speed, how much should it be?

3.   Figure 8 has four figures, it is better to add the corresponding icons. There are vague conclusions when describing the content of the picture.

The quality of English is good.

Author Response

Manuscript ID: bioengineering-2609759 rebuttal letter 

The authors are grateful for the comments received from the reviewer. All comments and suggestions have been considered and addressed as part of the revised version of the manuscript. Changes to the manuscript have been highlighted with a red font. 

Sincerely, 

Elias Sundström, Michael Jiang, Hani K. Najm, and Justin T. Tretter 
Stockholm, Sweden, 2023-10-03 

Reviewer #1 
This article presents the use of Blood Speckle Imaging (BSI) as a novel echocardiographic approach for the pre- and post-operative evaluation of subaortic membrane resection and aortic valve repair. The authors believe that this is the first paper describing the use of tissue and hemo-dynamics data with BSI. Sundström and co-authors sought to utilize BSI data in their clinical case to retrospectively analyze the flow patterns and derived properties to visually assess and quantify the hemodynamic changes from surgical subaortic membrane resection and aortic valvar repair. 

Major: 
1. The authors propose a new application scenario, but do not propose an innovation in principle. In particular, there is no specific principle, if necessary, please list specific formulas and specific detection mechanisms. 
A: We have used the commercial Vivid E95 ultrasound system and 9VT-D transesophageal probe by GE HealthCare (GE Vingmed Ultrasound AS, Horten, Norway). We are limited to investigate its implementation and we cannot list specific detection algorithms used with this system. 

2. Similar to laser Doppler and diffused light correlation spectroscopy measurement techniques, laser speckle contrast imaging methods are based on the basic principle of dynamic light scattering, and various approximate models are used to approximate the autocorrelation function of speckle to obtain the rate of moving scattering particles. Doppler technology is listed in the manuscript, but there is noadequately comparison, what are the advantages and disadvantages of BSI technology? 
A: This is a much appreciated remark. The relevant paragraphs in the Discussion section on the advantages and disadvantages of the BSI technology are now reworked for better clarity. 

3. The legends in Figures 7 and 8 are vague, and it is recommended to add a separate legend to each image. 
A: The legends to Figure 7 and 8 are rewritten for improved clarity. Separate legends for each subplot are also added. 

4. The structure of the manuscript is a bit strange, section 4 has only Discussion sections, and the Discussion section is too long. It should be divided into two sections, and a Conclusion section should be added. 
A: Thank you for this comment. The Discussion is now separated for improved clarity: 4. Discussion, 5. Limitations, and 6. Conclusions. 

Minor: 
1. Section 2.1 is too short and should be merged into other chapters. 
A: The authors appreciate this comment. This section is now extended. We feel that Section 2.1 belongs to a separate section instead of merging it with Sections 2.2 or 2.3. 

2. The speckle lining result correlates strongly with the flow rate of speckle particles per unit time, that is, the average velocity. In line 109, the peak speed is shown, if expressed in terms of average speed, how much should it be? 
A: We appreciate this remark. Since we do not obtain the 3D Velocity field from the CW Doppler data or BSI we cannot calculate the average speed via the surface intergral. However, we can integrate the BSI data from Figure 8 along the line profile, to get an approximate average speed in the 2D plane between the anterior and posterior endothelial walls. For the post-operative case at peak systole this integration gives an average speed of 0.9 m/s, which is lower than the max velocity 1.3 m/s. For the pre-operative case at peak systole we get an average speed of 0.6 m/s, which is 50\% lower than the max velocity 1.2 m/s. This result is not included since the aim is to make a corresponding comparison between the CW Doppler data and BSI. 

3. Figure 8 has four figures, it is better to add the corresponding icons. There are vague conclusions when describing the content of the picture. 
A: Separate legends are added for the subplots. Identified vague sentences have been rewritten for improved clarity.

Comments on the Quality of English Language:  
The quality of English is good. 

Reviewer 2 Report

The authors studied  Blood Speckle Imaging: an emerging method for perioperative evaluation of subaortic and aortic valvar repair

The article needs minor revision

1: The authors provided only the computational results, where as the mathematical model is avoided, therefore, the authors are suggested to add mathematical model as well.

2: The authors needs to add some studies relevent to the present analysis.

3: The graphical results needs more physical discussion.

4: The article needs to be revised against experimental data as well.

The language should be revised for language correction.

Author Response

Manuscript ID: bioengineering-2609759 rebuttal letter 

The authors are grateful for the comments received from the reviewer. All comments and suggestions have been considered and addressed as part of the revised version of the manuscript. Changes to the manuscript have been highlighted with a red font. 

Sincerely, 

Elias Sundström, Michael Jiang, Hani K. Najm, and Justin T. Tretter 
Stockholm, Sweden, 2023-10-03 

Reviewer #2 
The authors studied  Blood Speckle Imaging: an emerging method for perioperative evaluation of subaortic and aortic valvar repair 

The article needs minor revision: 
1: The authors provided only the computational results, where as the mathematical model is avoided, therefore, the authors are suggested to add mathematical model as well. 
A: We have used the commercial Vivid E95 ultrasound system and 9VT-D transesophageal probe by GE HealthCare (GE Vingmed Ultrasound AS, Horten, Norway). We are limited to investigate its implementation and we cannot list specific detection algorithms used with this system.  

2: The authors needs to add some studies relevent to the present analysis. 
A: We have added several studies that are relevant to the present analysis, e.g.: 
[6] Dockerill, C.; Gill, H.; Fernandes, J.F.; Nio, A.Q.; Rajani, R.; Lamata, P. Blood speckle imaging compared with conventional 290
Doppler ultrasound for transvalvular pressure drop estimation in an aortic flow phantom. Cardiovascular Ultrasound 2022, 291
20, 1–11.
[21] Stefani, L.; De Luca, A.; Maffulli, N.; Mercuri, R.; Innocenti, G.; Suliman, I.; Toncelli, L.; Vono, M.C.; Cappelli, B.; Pedri, S.; 323
et al. Speckle tracking for left ventricle performance in young athletes with bicuspid aortic valve and mild aortic regurgitation. 324
European Journal of Echocardiography 2009, 10, 527–531. https://doi.org/10.1093/ejechocard/jen332. 

3: The graphical results needs more physical discussion. 
A: The Discussion has been extended with more physical discussion to the graphical results. 

4: The article needs to be revised against experimental data as well. 
A: We make clear in the Limitation that there are caveat in this clinical study e.g. multilevel obstruction, and poor alignment of the TEE probe that may influence the result. It is further clarified in the Discussion that the technique needs validation. We make references to previous studies that have measured the pressure drop in silicon phantom models of the aortic valve using PendoTech pressure sensors to assess the accuracy of CW Doppler and BSI. However, so far it has been challenging to perform these pressure measurements in-vivo.

Comments on the Quality of English Language: 
The language should be revised for language correction. 
A: Identified sentences have been rewritten for improved clarity.

Reviewer 3 Report

It is always nice to read about methods that have immediate clinical applicability. A few clarifications and constructive suggestions, in no particular order of importance. Please bear in mind that these comments come from a clinician who is not an echo specialist.

1. Title. No need to capitalise the first 3 words, this can be done after defining the BSI abbreviation in the abstract. 

2. Pilot - presenting only one case does not exactly qualify for using this terminology, at least a handful of cases would be required. This is more like a case report or proof of concept study. It would be stronger if more than one case was presented here.

3. Velocity. The transesophageal prep examination shows a velocity of 3.5 m/s which is rightly classified as moderate. A velocity of 4 m/s is usually required to establish a surgical indication. One can assume that the TEE in this case did not achieve good alignment and that a higher gradient was confirmed by TTE in the run up to surgery; perhaps this can be clarified.

4. Next steps. Would it be possible to be a bit more specific about what could be done next to validate this technique. Please bear in mind that there are caveats in various clinical studies but validation work is nevertheless possible and ultimately a must. What I am trying to say is that in many of these cases there is multilevel obstruction. Furthermore, some aortic valves are palliated by surgery and it is accepted that a poor anatomic substrate can eventually lead to reoperation. The eventual reoperation does not necessarily mean that the previous procedure was inadequate or that any residual lesions were not properly assessed. These nuances can be better reflected in the discussion. If the final word count is too high I can see a number of places where judicious cuts can be made without loss of message. 

Author Response

Manuscript ID: bioengineering-2609759 rebuttal letter 

The authors are grateful for the comments received from the reviewer. All comments and suggestions have been considered and addressed as part of the revised version of the manuscript. Changes to the manuscript have been highlighted with a red font. 

Sincerely, 

Elias Sundström, Michael Jiang, Hani K. Najm, and Justin T. Tretter 
Stockholm, Sweden, 2023-10-03 

Reviewer #3 
It is always nice to read about methods that have immediate clinical applicability. A few clarifications and constructive suggestions, in no particular order of importance. Please bear in mind that these comments come from a clinician who is not an echo specialist. 

1. Title. No need to capitalise the first 3 words, this can be done after defining the BSI abbreviation in the abstract. 
A: Thank you. This has now been corrected.

2. Pilot - presenting only one case does not exactly qualify for using this terminology, at least a handful of cases would be required. This is more like a case report or proof of concept study. It would be stronger if more than one case was presented here. 
A: Thanks for this comment. The terminology using "pilot study" is now changed to "proof of concept study".

3. Velocity. The transesophageal prep examination shows a velocity of 3.5 m/s which is rightly classified as moderate. A velocity of 4 m/s is usually required to establish a surgical indication. One can assume that the TEE in this case did not achieve good alignment and that a higher gradient was confirmed by TTE in the run up to surgery; perhaps this can be clarified. 
A:  The surgical threshold for subaortic membrane is different than that for aortic valvar stenosis. This is especially true in children, with a common threshold to proceed with surgery being a peak instantaneous Doppler gradient greater than or equal to 50 mmHg (peak velocity ~3.5 m/s) or mean Doppler gradient >30 mmHg. This lower threshold for subaortic stenosis relates to identified risk factors for progression, such as reported in the study by Karamlous T et al. Prevalence and associated risk factors for intervention in 313 children with subaortic stenosis. Ann Thorac Surg 2007;84:900.  We agree with the reviewer on the assumptions regarding a higher pre-operative TTE gradient, related not only to the factors mentioned by the reviewer, but also the fact the pre-operative TEE is performed while the patient is intubated and under general anesthesia. The preceding TTE study for this patient had a peak gradient of 64 mmHg or peak velocity = 4.0 m/s, with a mean gradient of 35 mmHg, which was progressive compared to prior studies, with progression in the aortic regurgitation to mild to moderate. Since the current study is not meant to investigate surgical thresholds, but instead the use of BSI in pre and post-operative TEE for LVOT and aortic valve surgery, we would prefer not to delve into these details in the current manuscript, especially since even with the gradient achieved under general anesthesia by TEE the patient meets these general surgical thresholds for suboaortic membrane resection.

4. Next steps. Would it be possible to be a bit more specific about what could be done next to validate this technique. Please bear in mind that there are caveats in various clinical studies but validation work is nevertheless possible and ultimately a must. What I am trying to say is that in many of these cases there is multilevel obstruction. Furthermore, some aortic valves are palliated by surgery and it is accepted that a poor anatomic substrate can eventually lead to reoperation. The eventual reoperation does not necessarily mean that the previous procedure was inadequate or that any residual lesions were not properly assessed. These nuances can be better reflected in the discussion. If the final word count is too high I can see a number of places where judicious cuts can be made without loss of message. 
A:The author’s appreciate this comment. It is further clarified in the Discussion that the technique needs validation. We acknowledge at the end of the discussion that recurrence of the subaortic membrane may be inevitable in certain cases, meaning the need for reoperation does not necessarily indicate a failed prior surgery as the reviewer rightly points out. Lastly, we make references to previous studies that have measured the pressure drop in silicon phantom models of the aortic valve using PendoTech pressure sensors to assess the accuracy of CW Doppler and BSI. However, so far it has been challenging to perform these pressure measurements in-vivo.

Reviewer 4 Report

The use of Blood Speckle Imaging (BSI) as a cutting-edge echocardiographic technique for the pre- and post-operative assessment of subaortic membrane excision and aortic valve is presented in this study. The results demonstrated the viability of BSI and its potential therapeutic value in the clinical practice.

Comments:

1. Abstract seems like an advertisment. Please re-write it as background, aim, materials and methods, results and discussion.

More precise comparison of measured data with other methods is preferable.

2. Why one patient? 

3. Figures bigger.

4. More precise description of process how you measured the velocity.

Author Response

Manuscript ID: bioengineering-2609759 rebuttal letter 

The authors are grateful for the comments received from the reviewer. All comments and suggestions have been considered and addressed as part of the revised version of the manuscript. Changes to the manuscript have been highlighted with a red font. 

Sincerely, 

Elias Sundström, Michael Jiang, Hani K. Najm, and Justin T. Tretter 
Stockholm, Sweden, 2023-10-03 

Reviewer #4
The use of Blood Speckle Imaging (BSI) as a cutting-edge echocardiographic technique for the pre- and post-operative assessment of subaortic membrane excision and aortic valve is presented in this study. The results demonstrated the viability of BSI and its potential therapeutic value in the clinical practice.

Comments:

1. Abstract seems like an advertisment. Please re-write it as background, aim, materials and methods, results and discussion. More precise comparison of measured data with other methods is preferable.
A: The Abstract is now restructured for improved readability. Specific results of the comparison is now included.  

2. Why one patient? 
A: For this proof of concept study we only had consent from one patient undergoing subaortic membrane resection and aortic valve repair. In the Discussion we make clear that in the follow up study we will consider a larger cohort with both stenotic and regurgitant valves undergoing surgery.

3. Figures bigger.
A: Identified figures that seemed small have been made bigger by specifying width=\textwidth to fill all space in the single-column format.

4. More precise description of process how you measured the velocity.
A: We make clear that we have used the commercial Vivid E95 ultrasound system and 9VT-D transesophageal probe by GE HealthCare (GE Vingmed Ultrasound AS, Horten, Norway). We are limited to investigate its implementation and we cannot list specific detection algorithms used with this system to measure the velocity. However, we have made several references to other studies that describe similar detection algorithms in more detail. 

Round 2

Reviewer 1 Report

It has been revised according to the suggestion.  It is wished to do more testing if possible.

Reviewer 4 Report

Authors presented an updated version. We can accept it in present form.